# External Validation of the Hospital Frailty-Risk Score in Predicting Clinical Outcomes in Older Heart-Failure Patients in Australia

**DOI:** 10.3390/jcm11082193

**Published:** 2022-04-14

**Authors:** Yogesh Sharma, Chris Horwood, Paul Hakendorf, Rashmi Shahi, Campbell Thompson

**Affiliations:** 1College of Medicine and Public Health, Flinders University, Adelaide 5042, Australia; rashmi.shahi@flinders.edu.au; 2Department of General Medicine, Division of Medicine, Cardiac and Critical Care, Flinders Medical Centre, Adelaide 5042, Australia; 3Department of Clinical Epidemiology, Flinders Medical Centre, Adelaide 5042, Australia; chris.horwood@sa.gov.au (C.H.); paul.hakendorf@sa.gov.au (P.H.); 4Discipline of Medicine, The University of Adelaide, Adelaide 5005, Australia; campbell.thompson@adelaide.edu.au

**Keywords:** heart failure, frailty, hospital frailty-risk score, mortality, readmissions, length of hospital stay

## Abstract

Frailty is common in older hospitalised heart-failure (HF) patients but is not routinely assessed. The hospital frailty-risk score (HFRS) can be generated from administrative data, but it needs validation in Australian health-care settings. This study determined the HFRS scores at presentation to hospital in 5735 HF patients ≥ 75 years old, admitted over a period of 7 years, at two tertiary hospitals in Australia. Patients were classified into 3 frailty categories: HFRS < 5 (low risk), 5–15 (intermediate risk) and >15 (high risk). Multilevel multivariable regression analysis determined whether the HFRS predicts the following clinical outcomes: 30-day mortality, length of hospital stay (LOS) > 7 days, and 30-day readmissions; this was determined after adjustment for age, sex, Charlson index and socioeconomic status. The mean (SD) age was 76.1 (14.0) years, and 51.9% were female. When compared to the low-risk HFRS group, patients in the high-risk HFRS group had an increased risk of 30-day mortality and prolonged LOS (adjusted OR (aOR) 2.09; 95% CI 1.21–3.60) for 30-day mortality, and an aOR of 1.56 (95% CI 1.01–2.43) for prolonged LOS (c-statistics 0.730 and 0.682, respectively). Similarly, the 30-day readmission rate was significantly higher in the high-risk HFRS group when compared to the low-risk group (aOR 1.69; 95% CI 1.06–2.69; c-statistic = 0.643). The HFRS, derived at admission, can be used to predict ensuing clinical outcomes among older hospitalised HF patients.

## 1. Introduction

Worldwide, heart failure (HF) is a major cause for hospitalisation. In Australia, >150,000 patients are hospitalised with HF annually, with an estimated cost burden of AUD 2.3 billion [1]. Frailty is a multidimensional geriatric syndrome that is common in older hospitalised patients [2]. Frailty is characterised by decreased homeostatic reserves, which renders a person vulnerable to negative health outcomes, even from minor external stressors [3]. Recent studies [4,5] suggest that up to 50% of HF patients are frail; however, current risk prediction models in HF do not include frailty status. Frailty leads to adverse clinical outcomes in this population because it increases the risk of falls, disability and death [5,6,7]. The identification of frailty early in the course of hospital admission is of paramount importance so that appropriate measures can be applied—measures that can prevent the further deconditioning and worsening of frailty status [8].

Although a number of tools have been developed to identify frailty in the last two decades, there is still no universally accepted method to measure frailty in acute care settings, and it is often not measured [9]. Existing frailty tools may be time-consuming and, thus, may be impractical to use in volume-driven settings, such as acute care hospitals [10]. Another major limitation of the existing tools is that they require a face-to-face assessment by trained personnel and, sometimes, even require special equipment to identify frailty [11]. In addition, hospitalised patients may miss frailty assessments because of other reasons such as the busy work schedules of the clinicians, who may have to prioritise other assessments such as the measurement of vital signs [12]. Thus, frail patients may not be readily apparent to those health planners who may be interested in implementing quality improvement measures to improve their clinical outcomes. This issue can be mitigated by the development of a tool based on routinely collected hospital administrative data. One such tool, the hospital frailty-risk score (HFRS), has recently been validated within the UK National Health Service [13]. The HFRS is based upon administrative data and allocates point values for any of the 109 select ICD codes, as defined in the original publication. These codes include diagnoses such as falls, osteoporosis, spinal compression fractures, blindness, skin ulcers, delirium/dementia, Parkinson’s disease, urinary incontinence, urinary tract infections, disorders of electrolytes, drug/alcohol abuse and sequelae of stroke such as hemiplegia and dysphagia. None of the ICD-10 codes used for the generation of the HFRS score are for heart failure, atrial fibrillation or coronary artery disease (CAD) [9]. Increasing HFRS scores have been found to be associated with significantly increased risks of adverse health outcomes in older hospitalised patients [13]. This tool can, thus, be used to screen frail patients in a low-cost systematic way; however, it firstneeds validation against other established tools and in different health-care settings [13,14]. To date, this tool has not generally been applied using data available at the admission of a patient, nor data validated among HF patients, in Australia.

Therefore, the aims of this research are to determine whether the HFRS, calculated at admission, can be used to predict clinical outcomes in older hospitalised HF patients in Australian health-care settings.

## 2. Materials and Methods

### 2.1. Data Source 

This study was conducted at Flinders Medical Centre (FMC) and the Royal Adelaide Hospital (RAH), Adelaide, and included the data of all the HF patients who were referred from the Emergency Department (ED) for medical admission. We identified all the hospital admissions of adults ≥ 75 years, between 1 January 2013 and 31 December 2020, with a primary diagnosis of HF; this was achieved using the International Classification of Diseases Tenth Revision Australian Modification (ICD-10-AM), code 150, which has been previously used to define HF [15]. The protocol for this study was reviewed by the Southern Adelaide Human Research Ethics Committee (SA HREC) and was determined to be exempt, and the study was registered with the Australia and New Zealand Clinical Trial Registry ACTRN12622000013763.

We determined the HFRS by including only data from prior to the index admission, to determine whether the HFRS can be used to predict clinical outcomes among hospitalised HF patients at the time of hospital presentation. The HFRS was calculated from the international disease classification codes (ICD-10) that were documented for the participants’ previous admissions upon their discharge from hospital. This score was an aggregate of 109 ICD-10 diagnostic codes that were found by Graham et al. to correlate with frailty risk [13]. These diagnostic codes were awarded a specific value which was proportional to their strength of relationship with frailty. Higher scores were associated with a greater risk of frailty, and the scores were categorised into three levels as per the original study: low risk (<5 points), intermediate risk (5–15 points) and high risk (>15) [16]. 

Baseline comorbidities were assessed using the secondary diagnosis codes present during index hospitalisation, and they were used to calculate the Charlson comorbidity index [17]. Frailty status can be influenced by socioeconomic status, which was determined by use of the index of relative socioeconomic disadvantage (IRSD) [18]. The severity of HF was assessed using the N-terminal pro-brain natriuretic peptide levels (NT-proBNP) [19].

### 2.2. Clinical Outcomes

We determined the predictive ability of the HFRS on three clinical outcomes: mortality within 30-days of hospital admission, LOS > 7 days and unplanned readmissions within 30 days of hospital discharge. We determined the odds ratios and c-statistics to compare the results with other studies in order to validate the use of the HFRS in Australian health-care settings. 

The patient covariates considered included age, sex, the Charlson comorbidity index and the IRSD.

### 2.3. Statistics

Continuous data are presented as mean (SD) or median (IQR) and categorical data as proportions. Continuous variables were analysed using the t test and rank sum test as appropriate, and categorical variables using χ^2^ statistics.

To estimate the association of the HFRS with the outcomes, we used multilevel mixed logistic regression models with random effects to capture hospital level variation. We estimated the models first without adjustment, and then after adjustment for age, sex, Charlson index and the IRSD. The associations between the HFRS categories and clinical outcomes were evaluated using odds ratios, and c-statistics with 95% confidence intervals were used to test discrimination. We assessed model calibration by plotting the observed frequency of events per tenths of the predicted risk, as suggested by Altman et al. [20].

### 2.4. Sensitivity Analyses

We considered the HFRS as a continuous variable using splines and continuous forms to test its association with the clinical outcomes. In addition, we used Cox’s proportional hazard model to determine hazard ratios (HR) for mortality and readmissions. All statistical analyses were conducted using STATA software version 17.0. A *p* value of <0.05 was considered statistically significant.

### 2.5. Sample Size Calculation

The sample size for this study was based upon a previous study [21], which found that 30-day mortality in HFRS-defined-frail patients was 15.5% compared to 12.7% in non-frail patients. With an alpha level of 0.05 and a power of 80% a total sample size of 4844 was found to be sufficient.

## 3. Results

There were a total 8050 HF admissions between 2013–2020, of which the HFRSs of 5735 patients were available from previous admissions (Figure 1). The baseline characteristics of patients are shown in Table 1. 

The mean SD age was 76.1 (14.0) years, and 51.9% were females. The mean (SD) HFRS score was 3.1 (3.7), with a range of 0–25.8, and 797 (13.9%) patients were classified as frail. Patients with higher risks of frailty were older with a higher Charlson index and a greater severity of HF, as reflected by higher NT-proBNP levels (*p* < 0.05). Overall, 91 (1.6%) patients were on Sacubitril/Valsartan, and low-risk frail patients were significantly more likely to receive this medication when compared to those patients who were in the intermediate or high-risk frail categories (1.8% vs. 0.1% vs. 0, respectively, *p* < 0.001) (Table 1). The clinical outcomes in the three HFRS frail groups are presented in Table 2. 

The 30-day mortality was significantly higher in intermediate- and high-risk HFRS groups when compared to the low-risk HFRS category (Table 3: OR 2.02; 95% CI 1.60–2.54; and OR 3.11; 95% CI 1.85–5.22; c-statistic = 0.667). Likewise, there was an association between higher levels of HFRS-defined frailty and prolonged hospital LOS (OR 1.67; 95% CI 1.42–1.95; and OR 2.17; 95% CI 1.43–3.31; c-statistic 0.696) and 30-days readmissions (OR 1.64; 95% CI 1.37–1.96; and OR 1.79; 95% CI 1.13–2.84; c-statistic = 0.599. These trends remained significant after adjustment for covariates (adjusted OR 1.52; 95% CI 1.20–1.93 and 2.09; 95% CI 1.21–3.60; c-statistics = 0.730) for 30-day mortality, (adjusted OR 1.25; 95% CI 1.06–1.48 and 1.56; 95% CI 1.01–2.43; c-statistics 0.682) for prolonged LOS, and (adjusted OR 1.56; 95% CI 1.29–1.87 and 1.69; 95% CI 1.06–2.69; c-statistic = 0.643) for 30-day readmissions (Table 3). The model calibration was excellent for all models with a calibration slope around one, and the calibration line was close to the diagonal line (Figure 2).

The use of the HFRS as a continuous variable did not improve model discrimination (Table 4). The use of the survival analysis to assess 30-day mortality and readmissions generated Cox’s hazard ratios that were consistent with the results of the multilevel logistic regression analyses (Table 5). 

## 4. Discussion

Our results suggest that the HFRS can be used to predict imminent clinical outcomes in older hospitalised HF patients. The thresholds used to categorise patients into three frailty categories seemed adequate, and there was no additional value in using the HFRS as a continuous variable. The model calibration was found to be good for all three clinical outcomes.

HFRS has been previously validated in different countries and in specific populations. Our results were similar to other studies [4,22] with respect to the prediction of mortality and LOS, with similar c-statistics for mortality and LOS. The discrimination of our adjusted models for these clinical outcomes can be regarded as ‘good’ (c-statistics ~0.7 for both 30-day mortality and prolonged LOS), and matches existing studies of acute heart failure [23] and myocardial infarction [22]. 

In terms of 30-day readmissions, our results are similar to a US study [23] that included 348,619 hospitalisations for HF with a mean (SD) age of 80.1 (9.0) years, and found a significantly higher readmission risk among intermediate- (OR 2.53; 95% CI 2.43–2.63) and high-risk (OR 3.47; 95% CI 3.32–3.64) HFRS-defined-frail patients compared to those who were at low risk of frailty.

In contrast to our study, however, the above-mentioned studies [4,22,23] calculated the HFRS by including the data from each patient’s index hospitalisation, in addition to any other hospitalisations in the prior 2 years. The inclusion of index hospitalisation in the calculation of the HFRS dilutes the utility of this tool, because information in relation to a patient’s frailty status should ideally be available at the time of hospital admission, rather than at a later stage; this is so that frailty targeted interventions can be initiated early during admission when they can be most effective.

The HFRS can easily be determined from the ICD-10-AM codes, and has been validated using administrative data from different countries [13,16,24]. In addition, this tool has been validated against the two widely used clinical frailty scales—the Fried phenotype and the Rockwood Frailty Index—which require more time and additional resources for data collection [13]. Thus, the HFRS can be implemented without the need for patient assessment, and it can help to direct high-risk patients towards frailty-attuned interventions such as the Comprehensive Geriatric Assessment (CGA) [25]. In our study, nearly 14% of patients were at intermediate to high risk of frailty according to the HFRS. Thus, the HFRS can be used to identify those hospitalised HF patients who are at a high risk of short-term health-care use, and allow targeted strategies during and after admission; an example is the use of more intense follow-up care or post-acute care services during the vulnerable post-discharge period, to improve clinical outcomes such as mortality and readmissions.

The implementation of the HFRS in Australian health-care settings could represent a major advancement in the care of older HF patients with frailty, because it can alert (signpost) the authorities and can lead to a timely referral for goal-oriented, evidenced-based care—not just in retrospect but while the patient is in hospital and available for inpatient and early outpatient intervention. Two-thirds of patients >75 years old access acute hospital care more than once in two years [26], and those who have never accessed hospital care are typically at low risk of hospital-related adverse outcomes; thus the HFRS can especially be used to identify, at hospital presentation, those who are at a high risk of hospital-related adverse events and who are most likely to use considerable resources [27]. This tool might be useful to focus scarce hospital resources upon those vulnerable patients who are most likely to derive the most benefit from them.

### Strengths and Limitations

As the HFRS depends upon frailty-related coded information, it is possible that these codes are not adequately captured in hospitalised patients. This could be partly related to the inadequate documentation of frailty syndromes in case notes and discharge summaries by the treating medical teams. Furthermore, in hospitalised patients, a major emphasis is usually placed on major acute illnesses, which generate greater reimbursements for the hospital. Future studies can, thus, compare HFRS with an electronic frailty index that captures outpatient/ambulatory visits and is more likely to capture frailty-related codes. Further research can also explore the use of physiological parameters such as early warning scores at admission, and also explore the role of the polypharmacy in order to develop comprehensive frailty coding models.

## 5. Conclusions

In summary, our study supports the use, at admission, of the HFRS; it could potentially be used to identify high-risk older HF patients at admission who might require earlier and more intensive inpatient allied health services to maintain their functional status, if not improve their immediate clinical outcomes.

## Figures and Tables

**Figure 1 jcm-11-02193-f001:**
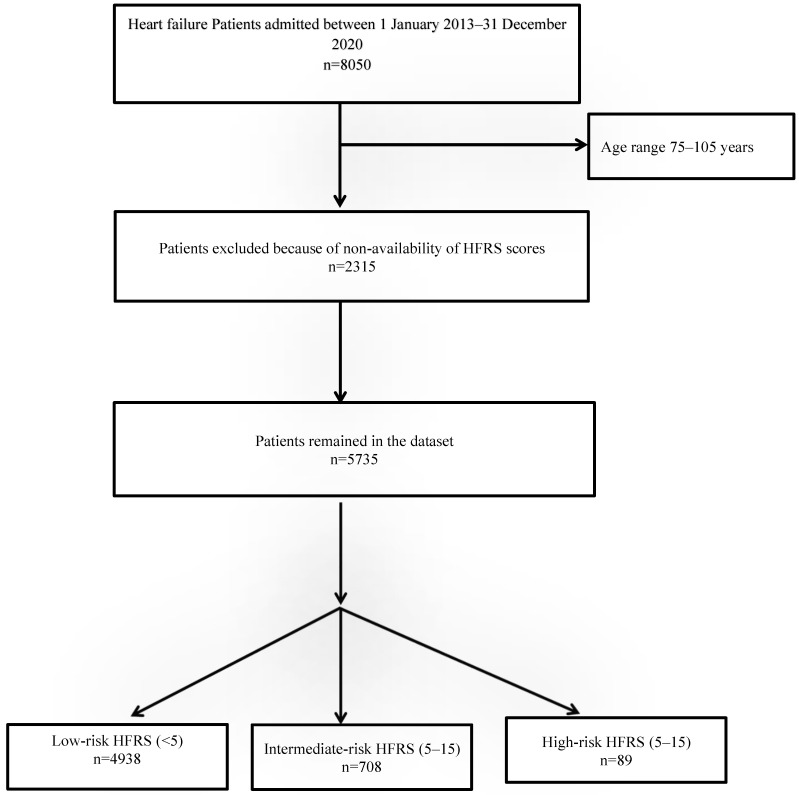
Study flow diagram.

**Figure 2 jcm-11-02193-f002:**
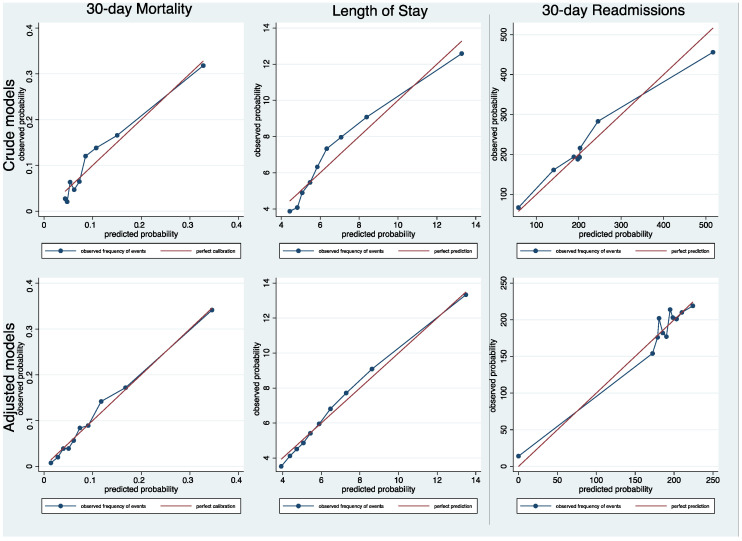
Calibration assessment after multilevel mixed logistic regression models for 30-day mortality, length of stay and 30-day readmissions.

**Table 1 jcm-11-02193-t001:** Baseline characteristics of patients.

	HFRS	Low Risk	Intermediate Risk	High Risk	*p* Value
	Overall	<5	5–15	>15	
Variable	*n* = 5735	*n* = 4938 (86.1%)	*n* = 708 (12.4%)	*n* = 89 (1.6%)	
Age mean (SD)	76.1 (14.0)	75.4 (14.3)	79.9 (11.3)	82.4 (9.6)	<0.001
Age group					
75–85 *n* (%)	3891 (67.8)	3436 (69.6)	412 (58.2)	43 (48.3)	<0.001
85–95	1671 (29.1)	1362 (27.6)	266 (37.6)	43 (48.3)	
>95	173 (3.1)	140 (2.8)	30 (4.2)	3 (3.4)	
Sex male *n* (%)	2975 (51.8)	2589 (52.4)	350 (49.4)	36 (40.6)	0.031
Charlson index mean (SD)	2.4 (1.7)	2.3 (1.6)	2.9 (1.9)	3.0 (2.2)	<0.001
Creatinine micromole/L mean (SD)	134.8 (75.9)	126.7 (70.8)	149.3 (68.1)	156.1 (84.8)	<0.001
Haemoglobin g/L *n* (%)	118.6 (19.4)	119.4 (19.2)	116.5 (20.1)	116.0 (18.7)	0.001
NT-proBNP ng/L mean (SD)	2052.7 (5575.5)	1682.9 (4720.7)	2969.9 (7192.3)	3781.8 (8382.7)	<0.001
IRSD mean (SD)	5.8 (2.6)	5.7 (2.6)	5.8 (2.5)	5.7 (2.9)	0.254
Sacubitril/Valsartan *n* (%)	91 (1.6)	90 (1.8)	1 (0.1)	0	<0.001
LOS median (IQR)	4.8 (2.8, 8.0)	5.4 (4.3, 9.1)	6.6 (4.8, 9.7)	8.7 (5.1, 10.0)	<0.001

HFRS—hospital frailty-risk score; SD—standard deviation; NT-proBNP—N-terminal pro-brain natriuretic peptide; IRSD—index of relative socioeconomic disadvantage, LOS—length of hospital stay; IQR—interquartile range.

**Table 2 jcm-11-02193-t002:** Clinical outcomes by HFRS category.

	HFRS	Low Risk	Intermediate Risk	High Risk	
	Overall	<5	5–15	>15	*p* Value
Variable	*n* = 5735	*n* = 4938	*n* = 708	*n* = 89	
30-day mortality	521 (9.1%)	396 (8.0%)	106 (14.9)	19 (21.3)	<0.001
LOS > 7 days	2233 (38.9%)	1832 (37.1%)	351 (49.6%)	50 (56.2%)	<0.001
30-day readmissions	3393 (20.8%)	908 (19.4%)	190 (29.4%)	25 (32.1%)	<0.001

HFRS—hospital frailty-risk score.

**Table 3 jcm-11-02193-t003:** Relationship between HFRS frailty risk and outcomes in 5182 heart-failure patients.

Outcome	Crude OR (95% CI)	Adjusted OR (95% CI)	*p* Value
30-day mortality			
Low risk (<5)	1.00	1.00	
Intermediate risk (5–15)	2.02 (1.60–2.54)	1.52 (1.20–1.93)	<0.001
High risk (>15)	3.11 (1.85–5.22	2.09 (1.21–3.60)	<0.001
c-statistic	0.667 (0.646–0.687)	0.730 (0.708–0.751)	
LOS >7 days			
Low risk (<5)	1.00	1.00	
Intermediate risk (5–15)	1.67 (1.42–1.95)	1.25 (1.06–1.48)	<0.001
High risk (>15)	2.17, (1.43–3.31)	1.56 (1.01–2.43)	<0.001
c-statistic	0.696 (0.676–0.715)	0.682 (0.667–0.696)	
30-day readmissions			
Low risk (<5)	1.00	1.00	
Intermediate risk (5–15)	1.64 (1.37–1.96)	1.56 (1.29–1.87)	<0.001
High risk (>15)	1.79 (1.13–2.84)	1.69 (1.06–2.69)	<0.05
c-statistic	0.599 (0.590–0.623)	0.643 (0.629–0.658)	

Multilevel mixed regression models were fitted with random effects to capture hospital level variation, first without adjustment, and then adjusted for age, sex, Charlson index and socioeconomic status. OR—odds ratio; CI—confidence interval; LOS—length of hospital stay.

**Table 4 jcm-11-02193-t004:** Prediction performance of continuous HFRS mixed models for outcomes. Models adjusted for age, sex, Charlson index and socioeconomic status.

Outcome	Crude Model c-Statistic with 95% CI	Adjusted Model c-Statistic with 95% CI
30-day mortality	0.620 (0.615–0.676)	0.731 (0.710–0.752)
Length of stay >7 days	0.592 (0.557–0.610)	0.682 (0.668–0.696)
30-day readmissions	0.560 (0.539–0.571)	0.578 (0.558–0.595)

CI—confidence interval.

**Table 5 jcm-11-02193-t005:** Cox proportional hazard risk models for association of HFRS risk with 30-day mortality and readmissions. Models adjusted for age, sex, Charlson index and socioeconomic status.

Outcome, HFRS Frailty Risk	Crude HR (95% CI)	Adjusted HR (95% CI)
30-day mortality		
Low risk (HFRS < 5)	1.00	1.00
Intermediate risk (HFRS 5–15)	2.30 (1.64–3.23)	1.74 (1.23–2.45)
High risk (HFRS > 15)	3.56 (1.74–7.26)	2.23 (1.10–4.64)
30-day readmissions		
Low risk (HFRS < 5)	1.00	1.00
Intermediate risk (HFRS 5–15)	1.61 (1.37–1.88)	1.52 (1.29–1.79)
High risk (HFRS > 15)	1.40 (1.31–2.14)	1.32 (1.11–2.01)

HFRS—hospital frailty-risk score; HR—hazard ratio; CI—confidence interval.

## Data Availability

The data presented in this study are available on request from the corresponding author, only after permission is granted by the ethics committee.

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
