# Peer review of "External Validation of the Hospital Frailty-Risk Score in Predicting Clinical Outcomes in Older Heart-Failure Patients in Australia"

_jcm, 2022, doi:10.3390/jcm11082193_

Round 1
Reviewer 1 Report
Congratulations on the very nice article. I would like to note one important thing the aim of the researchers was to determine whether the HFRS, calculated at admission, can be used to predict clinical outcomes in hospitalized HF patients in Australian health care settings. In my opinion, the study group of patients should be specified more precisely - for example as an older population because the authors included in their study only patients ≥75 years. The same clarification should be made in conclusions.
My second question is about natriuretic peptides level - did you measure only BNP level in study group? Why did you choose to measure BNP instead of NTproBNP? Were there any patients treated with sacubitryl/valsartan? That point should be clarified.
The p-value should be presented in Table 1, and the significant differences should be higlighted.
Author Response
Congratulations on the very nice article. I would like to note one important thing the aim of the researchers was to determine whether the HFRS, calculated at admission, can be used to predict clinical outcomes in hospitalized HF patients in Australian health care settings. In my opinion, the study group of patients should be specified more precisely - for example as an older population because the authors included in their study only patients ≥75 years. The same clarification should be made in conclusions.
Response: We thank reviewer for the comments and have now specified the study group in title, abstract, introduction, discussion, and conclusions sections of this manuscript.
My second question is about natriuretic peptides level - did you measure only BNP level in study group? Why did you choose to measure BNP instead of NTproBNP? Were there any patients treated with sacubitryl/valsartan? That point should be clarified.
Response: Our laboratory measures NT-proBNP levels and not BNP levels and this error has now been rectified in the text and Table 1.
Overall, 91 (1.6%) patients were treated with Sacubitril/Valsartan. Patients who were at low risk of frailty according to the HFRS, were significantly more likely to be on this medication compared to higher risk frailty categories ((1.8% vs. 0.1% vs. 0), P<0.001). We have now included this information in the results section on page 5 and Table 1.
“Overall, 91 (1.6%) patients were on Sacubitril/Valsartan and low risk frail patients were significantly more likely to receive this medication when compared to those patients who were in the intermediate or high risk frail categories (1.8% vs. 0.1% vs. 0, respectively, P<0.001)) (Table 1).”
The p-value should be presented in Table 1, and the significant differences should be higlighted.
Response: We have how presented the p values in Table 1.
Reviewer 2 Report
This is a very well done and well written article on Hospital Frailty Risk Score and outcomes in heart failure patients > 75 years. It's well known that HFRS predicts mortality and extended length of stay risks in older adults, but its performance in HF older patients is not proven.
I have some minor comments:
-The concept of frailty have to be extended in the introduction
- HFRS factors have to be reported in the paper.
Author Response
This is a very well done and well written article on Hospital Frailty Risk Score and outcomes in heart failure patients > 75 years. It's well known that HFRS predicts mortality and extended length of stay risks in older adults, but its performance in HF older patients is not proven.
Response: We thank reviewer for comments.
I have some minor comments:
-The concept of frailty have to be extended in the introduction
Response: We have expanded the concept of frailty in the introduction section as advised.
“Frailty is a multidimensional geriatric syndrome, which is common in older hospitalised patients [2]. Frailty is characterised by decreased homeostatic reserves, which renders a person vulnerable to negative health outcomes with even minor external stressors [3].”
- HFRS factors have to be reported in the paper.
Response: HFRS factors have now been reported as advised by the reviewer. Please refer to Introduction section page 2.
“HFRS is based upon administrative data by allocating point values for any of 109 select ICD codes as defined in the original publication. These codes include diagnoses such as falls, osteoporosis, spinal compression fractures, blindness, skin ulcers, delirium/dementia, Parkinson’s disease, urinary incontinence, urinary tract infections, disorders of electrolytes, drugs/alcohol abuse and sequelae of stroke such as hemiplegia and dysphagia. None of the ICD-10 codes used for the generation of the HFRS score is for heart failure, atrial fibrillation or coronary artery disease (CAD) [9].”